# Efficacy and safety of auricular therapy in the treatment of post-stroke constipation: A protocol for systematic review and meta-analysis

**Chunyu Ma**[1☯], **Ping Niu**[2☯], **Huifang Guan**[1], **Ziqiao Yu**[2], **Qiaoli Xu**[2], **Junchao Yu**[2], **Jing Su**[1], **Dexi Zhao** [2]*

1 Department of Traditional Chinese Medicine, Changchun University of Chinese Medicine, Changchun, Jilin Province, China, 2 Department of Neurology, First Affiliated Hospital of Changchun University of Chinese Medicine, Changchun, Jilin Province, China

☯ These authors contributed equally to this work.

* dexizhao1006@163.com

**Data Availability Statement:** No datasets were generated or analysed during the current study. Deidentified research data will be made publicly

## Abstract

### Background

Constipation is one of the common gastrointestinal complications after stroke. It not only aggravates the condition of stroke, but also brings huge medical burden to patients, and has a negative impact on the quality of life of patients. Auricular therapy, as a part of Chinese traditional acupuncture and moxibustion, has been found to be effective in the clinical treatment of constipation. However, no systematic review has investigated the efficacy and safety of auricular therapy in the treatment of post-stroke constipation. Therefore, the aim of this systematic review is to assess the effectiveness and safety of auricular therapy for post-stroke constipation.

### Methods and analysis

Eight electronic databases including PubMed, Cochrane Library/Cochrane Central Register of Controlled Trials, Web of Science, Embase, China National Knowledge Internet, Chinese Biomedical Literature Database, Wanfang, and VIP databases, will be searched for relevant studies published from inception to February 2023. Two reviewers will independently conduct research selection, data extraction, and evaluation of research quality. Only randomized controlled trials (RCTs) that assess the efficacy and safety of auricular therapy for the treatment of post-stroke constipation will be included in this study. We will use the Cochrane risk of bias assessment tool to evaluate the methodological qualities (including bias risk). If possible, a meta-analysis will be performed after screening.

### Results

This study may provide high-quality evidence for the efficacy and safety of auricular therapy in treating post-stroke constipation.

available when the study is completed and published.

**Funding:** This study was supported by Jilin Province Health Science and Technology Capacity Improvement Plan Project(NO.2021JC070) Changchun Science and Technology Development Project (NO.21ZGM32);Jilin Traditional Chinese Medicine Science and Technology Project in 2022 (NO.2022036).All funds conflict-free.

**Competing interests:** The authors declare that they have no known competing financial interests or personal relationships that could have appeared to influence the work reported in this paper.

**Abbreviations:** BSFS, Bristol stool form score; CI, confidence intervals; GRADE, Grading of Recommendations Assessment, Development and Evaluation; MD, mean difference; OR, odds ratios; PAC-QOL, patient assessment of constipation quality of life questionnaire; PRISMA, Preferred Reporting Items for Systematic Reviews and Meta-Analysis Protocol; RR, relative risk; SBMs, spontaneous bowel movements.

## Conclusion

The conclusions of our study will provide an evidence to judge whether auricular therapy is an effective and safe intervention for patients with post-stroke constipation.

## Ethics and dissemination

Ethical approval is not required, as this study was based on a review of published research. This review will be published in a peer-reviewed journal and disseminated electronically and in print.

## Trial registration

**Registration number:** PROSPERO CRD42023402242.

## 1. Introduction

In recent years, with the acceleration of social aging, as well as changes in lifestyle and other factors, the incidence rate of stroke has gradually increased, and the age of onset is younger [1]. It is reported that as one of the more common and very serious complications after stroke in clinical practice, the incidence of post-stroke constipation is 22.9%– 79% [2, 3]. Post-stroke constipation can cause varying degrees of harm to stroke patients, ranging from mild symptoms such as abdominal pain and bloating to severe symptoms such as intestinal obstruction due to constipation. In addition, potential hazards of constipation include: The long-term accumulation of feces in the intestinal tract can decompose endotoxin and be reabsorbed by the body, thereby aggravating the damage to other systems and even the central nervous system of the body; During forced defecation, the pressure in the abdominal cavity increases, which can induce hypertension and even lead to another stroke; Aggravate the brain nerve damage at the lesion site [4], affecting the prognosis of patients [5, 6]; Patients with post-stroke constipation often experience discomfort due to constipation symptoms, often accompanied by abnormal emotions such as anxiety and depression [7]. These seriously affect the quality of life of patients. Therefore, post-stroke constipation is a problem that needs attention and urgent solution. To explore an effective treatment plan for post-stroke constipation is not only an effective treatment for the symptom of constipation, but also has a promoting effect on the recovery of stroke and the cutoff of complications.

Clinically, commonly used methods for treating constipation include colon motility drugs, irritant/volumetric/permeable/softening/lubricating laxatives, or retention enemas. Although these drugs or external treatment methods can immediately and effectively excrete stool, it is difficult to maintain the efficacy after stopping the drug. Taking them for a long time or having toxic side effects on the body or forming drug dependence cannot effectively solve the problem of constipation in patients after stroke [8]. Traditional Chinese medicine technology has a unique and significant effect in the prevention and treatment of constipation symptoms after stroke. Whether it is oral administration of Chinese medicine or acupuncture and moxibustion therapy, from clinical literature reports, it can be seen that it has a good improvement effect on constipation after stroke, and does not have adverse reactions such as drug dependence. Therefore, searching for an effective treatment for post-stroke constipation from traditional therapies is a way to explore the treatment of post-stroke constipation in clinical practice.

Auricular therapy is an important part of Chinese traditional acupuncture, which refers to a traditional external treatment of Chinese medicine by sticking medicine beans or magnetic beads on ear points and pressing and stimulating ear points to make them feel sour, numb, swollen, etc. In recent decades, auricular therapy has been continuously improved while inheriting traditional auricular therapy, and has been widely applied in clinical disease treatment systems [9]. Therefore, this study intends to conduct a meta-analysis of the study of auricular therapy for post-stroke constipation, to explore its efficacy and safety, in order to provide reliable evidence for the clinical application of auricular therapy.

## 2. Methods

### 2.1. Study registration and ethics

The protocol of the present study adhered to the Preferred Reporting Items for Systematic Reviews and Meta-Analysis Protocol (S1 Appendix) guidelines and checklist [10] and the protocol is registered on PROSPERO under the registration number(CRD42023402242). The data for this study will be obtained from the published literature, and no ethical approval will be required for this systematic review.

### 2.2. Eligible criteria for study selection

**2.2.1. Types of studies.** All of the RCTs reporting the use of auricular therapy for the treatment of constipation in stroke patients should be included in this study. There are no restrictions on publishing status or language. Quasi-RCTs, non-RCTs, case reports, case series, non controlled trials, cross trials, reviews, observational study of clinical cases and animal studies will be excluded. If the studies fails to provide detailed results, it will also be excluded.

**2.2.2. Types of participants.** The subjects included in the study should meet various diagnostic criteria for cerebrovascular diseases. At the same time, symptoms of constipation should also be met. The diagnosis was based on the Rome II/III diagnostic criteria or the Chinese herbal medicine clinical research guide. There are no age, nationality, or race restrictions for the participants included in the study. However, clinical studies on subdural or subarachnoid hemorrhage, patients with a history of constipation or gastrointestinal disease or surgical history prior to stroke, auricular therapy as an adjuvant treatment, and studies with unclear diagnostic criteria will be excluded.

**2.2.3. Types of interventions.** Research using auricular therapy (Auricular therapy is a method for treating by stimulating a specific point in the ear, including medicine beans or magnetic beads, acupuncture, electroacupuncture, acupressure, drug injection, electric pulse stimulation, moxibustion, auricle bloodletting, etc.) or auricular therapy combined with other therapies(Western medicine, Chinese medicine/Chinese patent medicine, non-drug therapy, etc.). There are no restrictions on the acupoints and treatment courses in the study.

**2.2.4. Types of outcome measures.** For the primary outcome, we will assess the frequency of spontaneous defecation, defined as the mean number of spontaneous defecations per week. Secondary outcome measures include proportion of patients with defecation time, interval between bowel movements, number of bowel movements, changes in stool characteristics, etc. In addition, it is necessary to observe whether there are any adverse reactions during the treatment period.

### 2.3. Search strategies for the identification of studies

We will perform a comprehensive search using in PubMed, Cochrane Library/Cochrane Central Register of Controlled Trials, Web of Science, Embase, and four Chinese databases (China

**Table 1. The search strategy for PubMed.**

| Order | Strategy |
|---|---|
| #1 | Search: "Stroke"[Mesh] |
| #2 | Search: "Strokes"[Title/Abstract] OR "Cerebrovascular Accident"[Title/Abstract] OR "Brain Vascular Accident"[Title/Abstract] OR "Cerebrovascular Stroke"[Title/Abstract] OR "Apoplexy"[Title/Abstract] OR "Cerebral Stroke"[Title/Abstract] |
| #3 | #1 OR #2 |
| #4 | Search: "Constipation"[Mesh] |
| #5 | Search: "Constipation"[Title/Abstract] OR "Constipated"[Title/Abstract] OR "Constipating"[Title/Abstract] OR "Constipations"[Title/Abstract] |
| #6 | #4 OR #5 |
| #7 | Search: "auricular"[Title/Abstract] OR "auricular therapy"[Title/Abstract] OR "auricular acupuncture"[Title/Abstract] OR "auricular acupoints"[Title/Abstract] OR "auricular point sticking"[Title/Abstract] OR "electric pulse stimulation"[Title/Abstract] OR "auricular massage"[Title/Abstract] OR "auricular bleed"[Title/Abstract] |
| #8 | Search: "randomized controlled trial"[Publication Type] OR "RCT randomized controlled" [Publication Type] OR "random allocation"[Title/Abstract] OR "allocation,random" [Title/Abstract] OR "randomized, controlled"[Title/Abstract] OR "clinical trial" [Title/Abstract] |
| #9 | Search: "humans"[MeSH Terms] NOT "animals"[MeSH Terms] |
| #10 | #8 AND #9 |
| #11 | #3 AND #6 AND #7 AND #10 |

National Knowledge Internet, Chinese Biomedical Literature Database, Wanfang, and VIP databases). "Post-stroke constipation", "auricular therapy" and "Randomized clinical trials" as the key words or free-text terms. There are no restrictions of countries, population characteristics, language. The specific search strategies (for example, PubMed) are listed in Table 1. We will make relative modifications in accordance with the requirements, and an equivalent translation of the search terms will be adopted to ensure that similar search terms are used in all databases. If additional information is needed from the identified studies, we will contact the corresponding authors.

## 2.4. Data collection and analysis

### 2.4.1. Data extraction and management.

1. The literature search was performed independently by two authors (Chunyu Ma and Huifang Guan) through the database mentioned above.

2. The information of eligible study will be double-checked by two reviewers (Chunyu Ma and Huifang Guan) again. In case of disagreement on the suitability of a study for inclusion between the reviewers at this stage, the article will be checked by the other reviewer (Ziqiao Yu) again.

All studies identified through electronic and manual searches will be uploaded to EndNote 20 (Clarate Analytics). After deleting duplicates (the reasons for excluding the study will be recorded), a full text review of the included studies will be conducted. The above process is shown in the flowchart, as shown in Fig 1.

The information of screened studies will be extracted independently by two reviewers(Chunyu Ma and Huifang Guan). The data extraction content mainly includes:1. Basic information included in the research; 2. Baseline characteristics of the study subjects; 3. Specific details of interventions, comparison, follow-up time, etc.; 4. Key elements of risk assessment for bias; 5. Outcome indicators and outcome measurement data. In case of any disagreement on data

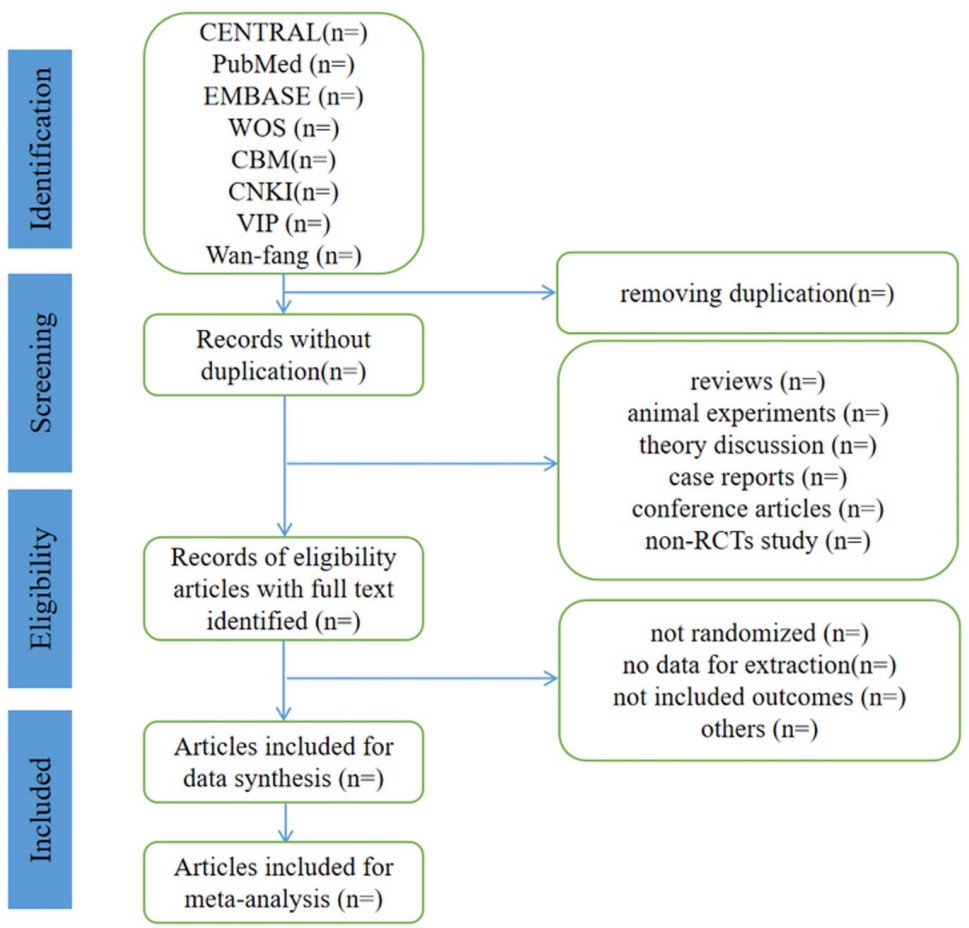

**Fig 1. Flow diagram of study selection process.**

extraction, it will be evaluated by the third reviewer (Ziqiao Yu). If there are unclear data, we will contact the corresponding author and gather relevant information.

**2.4.2. Assessment of bias risk and quality of included studies.** Two reviewers (Chunyu Ma and Huifang Guan) evaluated the bias risk of inclusion in the study according to the Cochrane Handbook 5. 1. 0 for RCT bias risk assessment tools [11], including random sequence generation, allocation concealment, blinding of patients and trial personnel, outcome assessors, outcome data completeness, selective reporting, other bias (declare fraud, potential bias related to the special study design, etc.). The risk of bias of the included literature was finally divided into three grades: "low", "high" and "unclear". If there is a disagreement, the discussion is resolved or judged by a third reviewer (Ziqiao Yu).

**2.4.3. Measurement of treatment effect.** Continuous variables were mainly measured by mean difference (MD) and 95% confidence intervals (CI), while dichotomous variables were measured by relative risk(RR) (or odds ratios(OR)) and 95%CI.

**2.4.4. Assessment of heterogeneity.** All statistical analyses will be performed by the Cochrane Collaboration Review Manager software (RevMan 5.3). $P<0.05$ indicates statistical significance. The heterogeneity among the results of the included literature was analyzed by $\chi^2$ test ($\alpha = 0.1$), and the heterogeneity was quantitatively determined by $I^2$. When $I^2$ 50%, it indicates no statistical heterogeneity or little heterogeneity, using fixed effects model for analysis; when $I^2 > 50\%$, analyze the source of heterogeneity, the subgroups of factors, analysis or

sensitivity analysis may cause heterogeneity analysis; if the included literature have statistical heterogeneity, sex without clinical heterogeneity, the random effects model was used for combined analysis. When 10 studies were included for an outcome measure, funnel plots were used to observe publication bias. We will use the Grading of Recommendations Assessment, Development and Evaluation (GRADE) pro software from Cochrane Systematic Reviews to create a Summary of Findings table.

**2.4.5. Subgroup analysis.** We will perform the following subgroup analysis to study heterogeneity when there is sufficient data. We will perform subgroup analysis based on age, gender, type of stroke (hemorrhagic and ischemic stroke), different definitions of constipation, stages of stroke (acute, subacute or chronic), and controls Type of group (placebo, no treatment or other active treatment or medication). A subgroup analysis based on different dietary groups takes into account that eating habits may play a role, and habits are playing an important role in the development of constipation. The intervention effect was analyzed using the $\chi^2$ test, with $p<0.05$, indicating a statistically significant difference between the subgroups.

**2.4.6. Sensitivity analysis.** We will perform a sensitivity analysis to verify the robustness of the results.

If the quality of the study is judged to be low (after evaluating the sample size, missing data, and the impact of the selected model), these studies will be deleted.

**2.4.7. Ethics and dissemination.** A formal ethical approval was not required for this protocol. The results of this review will be disseminated to peer-reviewed journals or presented at a relevant conference.

# 3. Discussion

Studies have shown that there is a strong correlation between brain function and the occurrence of constipation due to the interaction between the central nervous system and the intestinal autonomic nervous system [12]. During the onset and recovery of stroke, patients may experience abnormal intestinal activity, impaired anorectal and colonic sensory function, decreased control of internal and external sphincters, and subsequent neurogenic intestinal dysfunction due to factors such as the impact of stroke lesion sites on defecation in the central nervous system, long-term bed rest with hemiplegia, irregular defecation posture during recovery, and changes in dietary structure. This can lead to a high incidence of constipation after stroke. The treatment of post-stroke constipation in modern medicine cannot effectively solve the problem of post-stroke constipation in patients. In recent years, several clinical studies have reported that auricular therapy has shown good efficacy in the treatment of chronic constipation, and there is a trend to replace drugs in the treatment of chronic constipation. It has the advantages of safety, convenience, economy, small adverse reactions, and personalized treatment. Auricular therapy has the effect of treating constipation after stroke, and its mechanism may be related to stimulating the gastric region to help regulate the secretion of gastrointestinal hormones [13], improve vagal nerve excitability, promote colon smooth muscle movement, and improve gastrointestinal peristalsis [14]. A clinical study has shown that ear acupoint therapy can effectively promote the improvement of constipation symptoms after stroke, with faster results, especially in improving the time and frequency of first defecation [15].

When conducting systematic reviews and meta-analyses, the inclusion of RCTs is to minimize the impact of systemic bias. The design of RCTs can effectively reduce interference caused by other potential factors. This can more accurately evaluate the effectiveness of intervention measures. Meanwhile, the results of RCTs have wider applicability and can be promoted in different populations and trial conditions. Therefore, a systematic review and meta-

analysis containing RCTs can provide more reliable and consistent evidence, making the results of the systematic review and meta-analysis more credible and providing more reliable guidance for decision-makers and clinical practice. Although RCTs are important in systematic reviews and meta-analyses, we also need to recognize that other types of study designs have their unique advantages and application areas. Sometimes, when there are no RCTs available, it is also necessary to consider including non randomized studies for a systematic review and meta-analysis of intervention effects. For example, for diseases with a very regular disease process and significant intervention effects, there is no need to conduct RCTs, such as penicillin anti-infection treatment, as there are currently no RCTs with definite therapeutic effects. In certain surgical fields, conducting randomized trials for therapeutic or psychological interventions also violates ethical principles. When evaluating non randomized studies, attention should be paid to avoiding confounding bias and publication bias. The study of etiology or risk factors usually adopts an observational study design, such as case-control studies or cohort studies. When conducting systematic reviews and meta-analyses, researchers will comprehensively consider different types of research designs based on the characteristics and objectives of the research, and provide reasonable explanations and generalizations of the results. The more restrictions the evaluator has on the design type selected for a certain problem, the more limited the data they can collect. However, if the selected research cannot provide reliable information to answer the questions raised, such a systematic review may be useless and even lead to misleading conclusions. Therefore, our protocol is to only include RCTs for research.

Although multiple RCTs of auricular therapy as a treatment for post-stroke constipation have been reported to date, the cumulative evidence for its efficacy has not been systematically evaluated. This study will be the first systematic review of the efficacy of auricular therapy in patients with post-stroke constipation. In addition to providing evidence for the effectiveness and safety of the disease, this study will also analyze the limitations of research on auricular therapy for post-stroke constipation, such as the common issue of "researchers and journals should pay more attention to negative results", in order to provide reference for future clinical design and result analysis.

## Supporting information

**S1 Appendix. PRISMA-P (Preferred Reporting Items for Systematic review and Meta-Analysis Protocols) 2015 checklist: Recommended items to address in a systematic review protocol\*.**
(DOC)

## Author Contributions

**Conceptualization:** Chunyu Ma, Ping Niu, Dexi Zhao.

**Data curation:** Chunyu Ma, Huifang Guan.

**Formal analysis:** Chunyu Ma, Ziqiao Yu.

**Funding acquisition:** Qiaoli Xu, Dexi Zhao.

**Investigation:** Chunyu Ma, Qiaoli Xu.

**Writing – original draft:** Chunyu Ma, Junchao Yu, Jing Su.

**Writing – review & editing:** Chunyu Ma, Junchao Yu, Jing Su.

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
