## [Decision Letter · Decision Letter 0]

11 Dec 2023

PONE-D-23-23599Efficacy and safety of auricular therapy in the treatment of Post-stroke constipation:A protocol for systematic review and meta-analysisPLOS ONE

Dear Dr. Zhao,

Thank you for submitting your manuscript to PLOS ONE. After careful consideration, we feel that it has merit but does not fully meet PLOS ONE’s publication criteria as it currently stands. Therefore, we invite you to submit a revised version of the manuscript that addresses the points raised during the review process.

We look forward to receiving your revised manuscript.

Kind regards,

Md. Shahjalal

Academic Editor

PLOS ONE

Additional Editor Comments (if provided):

Please make sure the method section is clearer and specific. 

Reviewers' comments:

Reviewer's Responses to Questions

**Comments to the Author**

1. Does the manuscript provide a valid rationale for the proposed study, with clearly identified and justified research questions?

Reviewer #1: No

2. Is the protocol technically sound and planned in a manner that will lead to a meaningful outcome and allow testing the stated hypotheses?

Reviewer #1: No

3. Is the methodology feasible and described in sufficient detail to allow the work to be replicable?

Reviewer #1: No

4. Have the authors described where all data underlying the findings will be made available when the study is complete?

Reviewer #1: No

5. Is the manuscript presented in an intelligible fashion and written in standard English?

Reviewer #1: No

6. Review Comments to the Author

You may also provide optional suggestions and comments to authors that they might find helpful in planning their study.

Reviewer #1: This is a systematic review that aims to study the effectiveness and safety of auricular therapy for constipation of post-stroke. I have a few comments;

1. Why do you only include RCTs among the study designs?

2. Line 112: What is a non-standard Randomized controlled trial? Please correct R to r.

3. Line 114: Please correct Observational to observational.

4. Line 122: Why not include patients with subdural or subarachnoid hemorrhage?

5. 2.2.3. Types of interventions: Please describe auricular therapy in more detail.

6. 2.2.4. Types of outcome measures: I recommend that the primary outcome be an outcome commonly used in constipation studies.

7. 2.4.5 Please plan for subgroup analysis.

Overall, please double check spelling and spacing.

7. PLOS authors have the option to publish the peer review history of their article (what does this mean?). If published, this will include your full peer review and any attached files.

Reviewer #1: No

---

## [Author Response · Author response to Decision Letter 0]

27 Dec 2023

Dear Editors and Reviewers:

 We gratefully thank the editor and all reviewers for their time spend making their constructive remarks and useful suggestions, which has significantly raised the quality of the manuscript and has enable us to improve the manuscript. Each suggested revision and comment, brought forward by the reviewers was accurately incorporated and considered. Below the comments of the reviewers are response point by point and the revisions are indicated.

 Responds to the reviewer’s comments:

Reviewer #1:

1.Comment:Why do you only include RCTs among the study designs?

1.Reply:When conducting systematic reviews and meta-analyses, the inclusion of RCTs is to minimize the impact of systemic bias. The design of RCTs can effectively reduce interference caused by other potential factors. This can more accurately evaluate the effectiveness of intervention measures. Meanwhile, the results of RCTs have wider applicability and can be promoted in different populations and trial conditions. Therefore, a systematic review and meta-analysis containing RCTs can provide more reliable and consistent evidence, making the results of the systematic review and meta-analysis more credible and providing more reliable guidance for decision-makers and clinical practice.

 Although RCTs are important in systematic reviews and meta-analyses, we also need to recognize that other types of study designs have their unique advantages and application areas. Sometimes, when there are no RCTs available, it is also necessary to consider including non randomized studies for a systematic review and meta-analysis of intervention effects. For example, for diseases with a very regular disease process and significant intervention effects, there is no need to conduct RCTs, such as penicillin anti-infection treatment, as there are currently no RCTs with definite therapeutic effects. In certain surgical fields, conducting randomized trials for therapeutic or psychological interventions also violates ethical principles. When evaluating non randomized studies, attention should be paid to avoiding confounding bias and publication bias. The study of etiology or risk factors usually adopts an observational study design, such as case-control studies or cohort studies. When conducting systematic reviews and meta-analyses, researchers will comprehensively consider different types of research designs based on the characteristics and objectives of the research, and provide reasonable explanations and generalizations of the results.

 The more restrictions the evaluator has on the design type selected for a certain problem, the more limited the data they can collect. However, if the selected research cannot provide reliable information to answer the questions raised, such a systematic review may be useless and even lead to misleading conclusions. Therefore, our protocol is to only include RCTs for research. 

 At the same time, we will also include this part in the discussion of the paper to increase the completeness of the discussion. Please see page 12 of the revised manuscript, lines 239-249, and page 13, lines 250-266.

2.Comment:Line 112: What is a non-standard Randomized controlled trial? Please correct R to r.

2.Reply:We thank the reviewer for pointing this out. This is a typo. We have changed "Non standard randomized controlled trial" to "Quasi-RCTs" in the text. Please see page 5 of the revised manuscript, lines 110.

3.Comment:Line 114:Please correct Observational to observational.

3.Reply:We thank the reviewer for pointing this out. We have changed "Observational" to "observational" in the text. Please see page 6 of the revised manuscript, lines 111.

4.Comment:Line 122:Why not include patients with subdural or subarachnoid hemorrhage?

4.Reply:The commonly referred stroke refers to a condition caused by cerebral ischemia or cerebral hemorrhage. Subdural or subarachnoid hemorrhage is a special type of cerebrovascular disease, and its causes are different from those of stroke. Subarachnoid hemorrhage refers to the rupture of cerebral aneurysms, brain trauma, or other causes that lead to the rupture of cerebral blood vessels and the entry of blood into the subarachnoid space. In this case, blood can compress brain tissue and cause severe neurological symptoms. Subdural hemorrhage refers to the bleeding between the dura mater and the arachnoid membrane, which is commonly seen in the rupture of cortical arteries or veins at the site of cerebral contusion, or in the formation of cerebral hematoma penetrating the cortex and into the subdural space.

Due to the differences in pathogenesis and clinical manifestations between subdural or subarachnoid hemorrhage and other types of stroke, it may be necessary to study its related complications and sequelae separately, including constipation. Therefore, in systematic reviews and meta-analyses of post-stroke constipation, cerebral infarction and cerebral hemorrhage are usually the main focus, and cases of subdural or subarachnoid hemorrage are not included. This can make the research results more targeted and comparable.

5.Comment:2.2.3.Types of interventions: Please describe auricular therapy in more detail.

5.Reply:We thank the reviewer for pointing this out. Auricular therapy is a method for treating by stimulating a specific point in the ear, including medicine beans or magnetic beads, acupuncture, electroacupuncture, acupressure, drug injection, electric pulse stimulation, moxibustion, auricle bloodletting, etc. We have revised the text to address your concerns and hope that it is now clearer. Please see page 6 of the revised manuscript, lines 124-127.

6.Comment:2.2.4.Types of outcome measures: I recommend that the primary outcome be an outcome commonly used in constipation studies.

6.Reply:We thank the reviewer for pointing this out. For the primary outcome, we will assess the frequency of spontaneous defecation, defined as the mean number of spontaneous defecations per week. We have revised the text to address your concerns and hope that it is now clearer. Please see page 6 of the revised manuscript, lines 132, and page 7, lines 133.

7.Comment:2.4.5 Please plan for subgroup analysis.

7.Reply:We thank the reviewer for pointing this out. We will perform subgroup analysis based on age, gender, type of stroke (hemorrhagic and ischemic stroke), different definitions of constipation, stages of stroke (acute, subacute or chronic), and controls Type of group (placebo, no treatment or other active treatment or medication). A subgroup analysis based on different dietary groups takes into account that eating habits mayplay a role,and habits are playing an important role in the development of constipation. The intervention effect was analyzed using the χ2 test, with p<0.05, indicating a statistically significant difference between the subgroups. We have revised the text to address your concerns and hope that it is now clearer. Please see page 10 of the revised manuscript, lines 201-205, and page 11, lines 206-209.

At the same time, we have also made corresponding modifications to the spelling and spacing issues raised by the reviewers.

 We would like to thank the referee again for taking the time to review our manuscript.

---

## [Editor Report · Decision Letter 1]

26 Jan 2024

Efficacy and safety of auricular therapy in the treatment of Post-stroke constipation: A protocol for systematic review and meta-analysis

PONE-D-23-23599R1

Dear Dr. Dexi Zhao,

We’re pleased to inform you that your manuscript has been judged scientifically suitable for publication and will be formally accepted for publication once it meets all outstanding technical requirements.

Kind regards,

Md. Shahjalal

Academic Editor

PLOS ONE

Additional Editor Comments (optional):

Dear authors,

Thank you for pint by point response.

Please make sure same reference style and PLoS ONE guidelines.

Wish you the best.
---

## [Editor Report · Acceptance letter]

16 Feb 2024

PONE-D-23-23599R1 

PLOS ONE

Dear Dr. Zhao, 

I'm pleased to inform you that your manuscript has been deemed suitable for publication in PLOS ONE. Congratulations! Your manuscript is now being handed over to our production team.

Kind regards, 

on behalf of

Dr. Md. Shahjalal 

Academic Editor

PLOS ONE